# Benchmarking Message Queues

Rokin Maharjan [ORCID], Md Showkat Hossain Chy, Muhammad Ashfakur Arju and Tomas Cerny *[ORCID]

Department of Computer Science, Baylor University, Waco, TX 76706, USA; rokin_maharjan1@baylor.edu (R.M.)
* Correspondence: tomas_cerny@baylor.edu

**Abstract:** Message queues are a way for different software components or applications to communicate with each other asynchronously by passing messages through a shared buffer. This allows a sender to send a message without needing to wait for an immediate response from the receiver, which can help to improve the system's performance, reduce latency, and allow components to operate independently. In this paper, we compared and evaluated the performance of four popular message queues: Redis, ActiveMQ Artemis, RabbitMQ, and Apache Kafka. The aim of this study was to provide insights into the strengths and weaknesses of each technology and to help practitioners choose the most appropriate solution for their use case. We primarily evaluated each technology in terms of latency and throughput. Our experiments were conducted using a diverse array of workloads to test the message queues under various scenarios. This enables practitioners to evaluate the performance of the systems and choose the one that best meets their needs. The results show that each technology has its own pros and cons. Specifically, Redis performed the best in terms of latency, whereas Kafka significantly outperformed the other three technologies in terms of throughput. The optimal choice depends on the specific requirements of the use case. This paper presents valuable insights for practitioners and researchers working with message queues. Furthermore, the results of our experiments are provided in JSON format as a supplement to this paper.

**Keywords:** message queue; Redis; ActiveMQ Artemis; RabbitMQ; Apache Kafka; events; latency; throughput

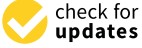



## 1. Introduction

In the ever-evolving landscape of distributed systems, the demand for robust, scalable, and asynchronous communication among their intricate components is paramount. Asynchronous messaging allows system elements to interact without the immediate need for responses, empowering components to dispatch messages and proceed with other tasks concurrently. This not only enhances the overall system throughput but also reduces response times, enabling systems to tackle complex workloads effectively and deliver real-time results [1]. At the heart of distributed computing lie message queues, pivotal in providing such capabilities. By decoupling services and fostering resiliency, fault tolerance, and responsiveness, message queues play a vital role in modernizing and optimizing distributed systems. Nonetheless, selecting the optimal message queue remains a daunting challenge.

The vast array of available options, each with its own unique strengths and weaknesses, necessitates a thorough understanding of these systems to ensure the performance and efficiency of the overall architecture. Choosing the right message queue is critical for cost optimization, improved system performance, versatility, and scalability. It enables significant cost savings in infrastructure and operational expenses while enhancing system responsiveness through reduced latency, high throughput, and timely message delivery. Message queues with advanced features facilitate the implementation of complex messaging patterns, workflow orchestration, and seamless integration of components.

This manuscript delves into a comprehensive benchmarking study that meticulously compares and evaluates four popular message queues: Redis, ActiveMQ Artemis, Rab-

bitMQ, and Apache Kafka. These message queues represent leading solutions in the field, each with its own distinctive characteristics and strengths. Redis, widely recognized as an open-source in-memory data store, not only excels in its primary role but also serves as a versatile and efficient message broker. ActiveMQ Artemis, specifically designed to cater to enterprise-level applications, offers high-performance and reliable messaging solutions, ensuring seamless communication in demanding and complex environments. Lastly, RabbitMQ, with its extensive developer community and broad adoption, stands as a robust and feature-rich message broker, supporting multiple messaging protocols and providing a solid foundation for scalable and flexible communication infrastructures.

The goal of this study was to equip developers and system architects with valuable insights for making informed decisions when selecting a message queue that aligns with their unique use cases. It is essential to note that a message queue that excels in one scenario might falter in another. To address this challenge, we meticulously designed a series of experiments encompassing a diverse spectrum of workloads and scenarios. Our evaluation revolves around two fundamental performance metrics: latency and throughput. Latency measures the time taken for a message to traverse from sender to receiver, whereas throughput quantifies the number of messages processed within a given timeframe. By thoroughly examining the performance of each message queue across various conditions, we gain comprehensive insights into their respective strengths and weaknesses, empowering us to make well-informed decisions. Moreover, the findings of this study contribute to the evolution and optimization of message queue systems at large, highlighting potential areas for future development and refinement.

To conduct our experiments, we utilized the OpenMessaging Benchmark Framework [2], a performance testing tool created by The Linux Foundation. This tool was specifically designed to measure the performance of messaging systems across various workloads and scenarios. Notably, it supports multiple messaging protocols and message queues, providing a unified testing framework for comprehensive evaluation. The tool offers detailed metrics for latency and throughput, allowing for precise performance analysis.

Our choice of this benchmarking tool was based on several compelling reasons. Firstly, it allowed us to maintain consistency by supporting the benchmarking of all four message queues in our study. This ensured that the evaluation process was unbiased and devoid of any favoritism toward specific technologies. Secondly, the OpenMessaging Benchmark Framework was developed by The Linux Foundation, a widely respected organization known for its contributions to open-source technologies. This factor ensured the tool's reliability and credibility. Lastly, the popularity of the tool among developers was evident from its impressive statistics on GitHub, including 295 stars, 183 forks, and contributions from 47 contributors at the time of writing.

The remaining sections of this manuscript are structured as follows: Section 2 provides an introduction to the four message queues that we benchmarked. Section 3 discusses related works and how they compare to our study. The methodology that we used to benchmark the message queues is presented in Section 4. Sections 5 and 6 describe the experiments conducted and their results. In Section 7, we discuss our results. Lastly, Section 8 summarizes our findings.

## 2. Background

In this section, we introduce the four message queues that are evaluated in this manuscript. We selected ActiveMQ Artemis, RabbitMQ, and Apache Kafka because they are three of the most popular message queues. We chose Redis because although Redis can be used as a message broker, it is not primarily used for that. We wanted to see how Redis would perform as a message broker because, if it performed well, it could be used to serve multiple purposes.

### 2.1. Redis [3]

Redis is a widely used in-memory data store that can act as a cache or a database. However, it also has the capability to serve as a message broker, which is a crucial component in message-driven architectures. A message broker serves as a central hub for receiving, storing, and forwarding messages between producers and consumers. By implementing the publish–subscribe pattern through its Pub/Sub feature, Redis can act as a message broker.

In Redis Pub/Sub, a publisher sends a message to a specific channel, and all the subscribers who have subscribed to that channel will receive the message. Clients can subscribe to one or more channels and receive messages whenever they are published to those channels. This enables messages to be broadcasted to a large number of subscribers in real-time, allowing for efficient and scalable communication between producers and consumers.

To implement Redis as a message broker, one can simply use the PUB/SUB commands to send and receive messages. Publishers can send messages to specific channels using the PUBLISH command, while subscribers can listen for messages by subscribing to specific channels using the SUBSCRIBE command. Redis clients can also be configured to listen for messages using the message callback function that is triggered when a message is received.

### 2.2. ActiveMQ Artemis [4]

ActiveMQ Artemis is an open-source message broker that can be used to send, receive, and store messages. It provides a messaging system that is designed to be scalable and high-performing and can be deployed in a variety of environments, including standalone applications, microservices, and cloud-based systems.

ActiveMQ Artemis is built using a modular architecture that allows it to be easily extended and customized to meet specific messaging requirements. It is written in Java and supports multiple messaging protocols, including AMQP (Advanced Message Queueing Protocol) [5], MQTT (Message Queuing Telemetry Transport) [6], STOMP (Simple Text Oriented Messaging Protocol) [7], and OpenWire [8].

As a message broker, ActiveMQ Artemis acts as an intermediary between producers and consumers of messages. It receives messages from producers and stores them until they can be delivered to the appropriate consumers. This allows applications to communicate with each other in a decoupled manner without the need for direct point-to-point communication.

ActiveMQ Artemis supports several messaging patterns, including point-to-point and publish–subscribe. Point-to-point messaging involves sending a message to a specific destination, such as a queue, whereas publish–subscribe messaging involves sending a message to multiple destinations, such as topics. ActiveMQ Artemis also provides support for advanced messaging features, including durable subscriptions, message grouping, and message routing. It also supports clustering, which allows multiple brokers to be connected together to form a single, highly available messaging system.

### 2.3. RabbitMQ [9]

RabbitMQ is a popular open-source message broker that is widely used in distributed systems. It is written in the Erlang programming language and is based on the Advanced Message Queuing Protocol (AMQP) standard.

As a message broker, RabbitMQ acts as an intermediary between applications that need to send and receive messages. Producers send messages to RabbitMQ, which stores them until they are consumed by consumers. This decouples producers and consumers, allowing them to operate independently of each other and ensuring reliable message delivery even if one of the systems is down.

RabbitMQ supports a variety of messaging patterns, including point-to-point, publish–subscribe, and request–response. It also provides advanced features such as message routing, message acknowledgments, and dead-letter queues.

RabbitMQ uses exchanges and queues to route messages between producers and consumers. An exchange receives messages from producers and routes them to one or more queues based on a set of rules called bindings. Consumers then subscribe to specific queues to receive messages.

*2.4. Apache Kafka [10]*

Apache Kafka is an open-source distributed streaming platform that is designed for high-throughput, fault-tolerant, and scalable real-time data streaming. It provides a messaging system that allows applications to publish, subscribe, and process streams of records in a fault-tolerant and distributed manner.

At its core, Kafka is based on a publish–subscribe model, where producers send records to topics and consumers subscribe to those topics to receive the records. Kafka stores the records in a distributed and fault-tolerant manner, enabling them to persist and be replicated across multiple nodes or clusters.

One of the key features of Kafka is its ability to handle large volumes of data and a high message throughput. It is known for its horizontal scalability, allowing it to handle millions of messages per second. Kafka achieves this high throughput by leveraging a distributed architecture that partitions data across multiple brokers and allows for parallel processing.

Kafka uses topics to categorize and organize messages. Producers publish records on specific topics, and consumers can subscribe to one or more topics to consume the records. Topics can be divided into multiple partitions, which allows for parallel processing and enables high scalability. Additionally, Kafka retains messages for a configurable period of time, allowing consumers to consume messages at their own pace.

## 3. Related Works

In the realm of benchmarking message queue systems, several studies have contributed valuable insights into their performance characteristics. Piyush Maheshwari and Michael Pang [11] conducted a benchmarking study comparing Tibco Rendezvous [12] and Progress Sonic MQ [13] using SPECjms2007 [14], focusing on evaluating message delivery latency, throughput, program stability, and resource utilization. While their study provided insights into specific message-oriented middleware (MOM), it differs from our research by not encompassing the comparative analysis of Redis, ActiveMQ Artemis, RabbitMQ, and Apache Kafka. While informative for specific message-oriented middleware (MOM), their study differs from ours as we compare Redis, ActiveMQ Artemis, RabbitMQ, and Apache Kafka, providing a broader comparative analysis.

In a related study, Kai Sachs et al. [15] conducted a performance analysis of Apache ActiveMQ using SPEC JMS 2007 [14] and jms2009-PS [16], comparing its usage as a persistence medium with databases. In contrast, our research expanded the scope to include Redis, ActiveMQ Artemis, RabbitMQ, and Apache Kafka, providing a comprehensive evaluation of these message queue technologies. We extensively assessed their latency and throughput performance across a diverse range of workloads, enabling practitioners to make informed decisions based on specific use cases. Furthermore, Stefan Appel et al. [17] proposed a unique approach for benchmarking AMQP implementations, whereas our study focused on a comparative analysis of different message queue technologies, offering valuable insights into their respective strengths and weaknesses.

Due to the closed-source nature of SPECjms2007 and jms2009-PS, several benchmarking solutions have been developed to provide a fair comparison between message queues, with the OpenMessaging Benchmark Framework standing out as a notable choice. Souza et al. [18] focused on evaluating the performance of Apache Kafka and RabbitMQ in terms of throughput, latency, and resource utilization using the OpenMessaging Benchmark (OMB) tool. Their findings reveal that Apache Kafka outperformed RabbitMQ in terms of throughput and scalability, particularly under heavy workloads with large messages. Additionally, RabbitMQ showcased lower latency and resource utilization, suggesting its

suitability for low-latency and resource-constrained environments. However, our study further performed a comparative analysis by incorporating Redis and ActiveMQ Artemis alongside Apache Kafka and RabbitMQ. Through comprehensive evaluations of latency and throughput across diverse workloads, we aim to provide practitioners with valuable insights into the strengths and weaknesses of these four message queue technologies, facilitating informed decision-making in choosing the most suitable solution for their specific needs.

Fu et al. [19] proposed a framework used to compare the performance of popular message queue technologies, including Kafka, RabbitMQ, RocketMQ, ActiveMQ, and Apache Pulsar [20]. Their research focused on evaluating factors such as message size, the number of producers and consumers, and the number of partitions. The study highlighted Kafka's high throughput due to optimization techniques but noted its latency limitations with larger message sizes. In our study, we specifically examined Redis, ActiveMQ Artemis, RabbitMQ, and Apache Kafka, providing a comparative analysis of their performance across diverse workloads.

John et al. [21] conducted a performance comparison between Apache Kafka [10] and RabbitMQ in terms of throughput and latency. Their study explored scenarios involving single and multiple publishers and consumers using the Flotilla [22] benchmarking tool. The results indicate that Kafka exhibited a superior throughput, whereas RabbitMQ prioritized reliability, especially in scenarios where data security was crucial. Our research is more extended since we considered including Redis and ActiveMQ Artemis and assessing the performance of these message queues under various workloads and scenarios.

Valeriu Manuel Ionescu et al. [23] conducted an analysis of RabbitMQ and ActiveMQ, specifically focusing on their publishing and subscribing rates. Their study employed different-sized images as a real-world comparison instead of traditional byte string loads and considered both single and multiple publisher–consumer scenarios. While their research highlighted performance differences between RabbitMQ and ActiveMQ, our study extended the comparative analysis to include Redis and Apache Kafka. Additionally, we evaluated the latency and throughput of these message queues, presenting more detailed results, including percentile-based end-to-end latency metrics.

Marko et al. [24] conducted a study focusing on message queueing technologies for flow control and load balancing in the IoT scenario. They specifically evaluated RabbitMQ and Apache Kafka within a smart home system cloud, assessing their performance with different numbers of consumers. The results highlight that Kafka exhibited stable data buffering and a lower average CPU usage, with no instances of reaching maximum CPU usage during testing. In comparison, our work extended beyond their scope by examining additional message queue technologies, including Redis and ActiveMQ Artemis. Furthermore, we provided a comprehensive analysis of latency and throughput across a diverse range of workloads.

Our study examined Redis, ActiveMQ Artemis, RabbitMQ, and Apache Kafka, shedding light on their respective performance characteristics. We assessed Redis's publish/subscribe operations and evaluated the enhanced ActiveMQ Artemis rather than the traditional version. Notably, our findings highlight ActiveMQ Artemis' advantageous latency performance in scenarios with low throughput, distinguishing it from RabbitMQ. Additionally, we provided comprehensive results featuring distinct graphs for throughput and latency, encompassing various percentiles. To ensure unbiased and consistent results, we utilized the OpenMessaging Benchmark tool from The Linux Foundation, a trusted and popular open-source solution.

The comparison between our study versus the existing studies is shown in Table 1.

**Table 1.** Comparison: our study vs. existing studies.

| References | Message Queues | | | | | Metrics | | | |
|---|---|---|---|---|---|---|---|---|---|
| | Redis | Kafka | Active MQ | RabbitMQ | Pulsar | Latency | Throughput | Persistence | Other Metric |
| Piyush et al. [11] | | | | | | | | | ✓ |
| Sachs et al. [15] | | | ✓ | | | | | ✓ | |
| Souza et al. [18] | | ✓ | | ✓ | | ✓ | ✓ | | ✓ |
| Fu et al. [19] | ✓ | ✓ | ✓ | ✓ | ✓ | ✓ | ✓ | | |
| John et al. [21] | | ✓ | ✓ | | | ✓ | ✓ | | |
| Valeriu et al. [23] | | | ✓ | ✓ | | | | | ✓ |
| Marko et al. [24] | | ✓ | | ✓ | | | | | ✓ |
| Our study | ✓ | ✓ | ✓ | ✓ | | ✓ | ✓ | | |

## 4. Methodology

This section describes the two metrics that we considered and why they play a crucial role in choosing a message queue. This section also outlines the process of installing the technologies, creating the experiments, and running them.

The two metrics considered in this study were latency and throughput. They are critical metrics to consider when benchmarking message queues because they provide insights into the system's performance and reliability. Latency refers to the time it takes for a message to be sent from the producer to the consumer. It is a measure of the system's responsiveness and can have a significant impact on user experience. High latency can cause delays and make the system feel sluggish, whereas low latency means that messages are delivered quickly, which can enhance the user experience. Throughput, on the other hand, is the number of messages a system can process over a specified period of time. It is a measure of the system's efficiency and capacity. A high throughput means that the system can handle a large volume of messages efficiently, whereas a low throughput indicates that the system may struggle to keep up with the demands placed on it.

The end-to-end latency graphs are typically used to understand the amount of time it takes for a request to be processed from the time it is initiated to the time it is completed. We represent the end-to-end latency in four different forms: 50th percentile, 75th percentile, 95th percentile, and 99th percentile latency. The 50th percentile (also known as the median) represents the maximum latency for the fastest 50% of all requests to complete. For instance, if the 50th percentile latency is 1 s, then the system processed 50% of requests in less than 1 s. Similarly, the 75th percentile, 95th percentile, and 99th percentile represent the time it takes for the fastest 75%, 95%, and 99% of all requests, respectively, to complete. When one has end-to-end latency graphs for these percentiles, one can use them to understand the overall performance of their system. If the latency for the 50th percentile is low, it means that the majority of requests are completed quickly. However, if the latency for the 99th percentile is high, it means that a small percentage of requests are taking a very long time to complete, and this could be a sign of a bottleneck or issue that needs to be addressed. Additionally, if one notices that the latency for the 75th percentile is increasing over time, it could be an indication that their system is becoming overloaded and needs to be scaled up.

We represent the throughput data in two different forms: in terms of megabytes per second (MBps) and the number of events per second. The MBps metric refers to the amount of data that can be transferred per second, and it provides a measure of the message queue's overall network performance. This metric is particularly useful in scenarios where the size of the data being transferred is large, such as when transferring multimedia files or large datasets. It also helps to measure the efficiency of network bandwidth utilization, and it can help to identify bottlenecks or limitations in the network infrastructure. On the other hand, measuring throughput in terms of the number of events per second provides insights into the message queue's performance in processing and delivering messages. This metric is particularly relevant in applications where real-time data processing and low latency are critical. For instance, in high-frequency trading systems or real-time data streaming applications, the number of events processed per second is a crucial factor in

determining the system's overall performance. Therefore, having throughput data in two different forms provides a more comprehensive view of the message queue's performance, taking into account both network bandwidth and message processing efficiency. This can help developers, system architects, and decision-makers to make more informed decisions when selecting or optimizing message queues for specific use cases.

For the purpose of running the experiments, all four message queues were installed locally following the official documentation. We used Redis server 7.0.9 [25], RabbitMQ 3.11.10 [26], ActiveMQ Artemis 2.28.0 [27], and Apache Kafka 3.4.0 [28]. We forked [29] the OpenMessaging Benchmark to customize the workloads (data size and the number of messages per second) according to the needs of our experiments.

In order to benchmark the four message queues, we used the OpenMessaging Benchmark tool created by The Linux Foundation. It is a highly versatile benchmarking tool written in Java that supports multiple popular message queues and data streaming technologies such as RabbitMQ, ActiveMQ Artemis, Redis, Apache Kafka, Pravega, Pulsar, and many more. It provides a range of pre-built payloads of sizes 100 B, 200 B, 400 B, 1 KB, 2 KB, and 4 KB. Besides these, it also provides the ability to create payloads of other sizes programmatically. Using this tool, we generated detailed metrics for latency and throughput, which are the metrics that we used to compare the four message queues. We used this tool because we wanted a common tool to benchmark the four technologies so as to maintain consistency and eliminate any biases that may arise if we had used benchmarking tools specific to each technology.

Using the feature provided by OpenMessaging, we generated data of various sizes. The specifics of how these data sizes were used and how the experiments were conducted are explained in detail in Section 5 (Experiments). After conducting multiple sets of experiments for both throughput and latency, we generated JSON [30] documents containing the results of the experiments. We built Python [31] scripts using the Matplotlib [32] library in order to represent the JSON documents in charts. The scripts can be found under */bin/charts* in our forked version [29] of OpenMessaging.

We adopted this benchmarking approach for the message queues in our study due to several compelling reasons. Firstly, latency and throughput are crucial metrics that determine the performance of a message queue, making them essential aspects to evaluate. Secondly, the OpenMessaging Benchmark tool, developed by The Linux Foundation, was selected for its credibility in the field of computer science. By utilizing the same tool, we ensure consistent and reliable benchmarking across the four message queues under examination. This approach is vital as using different tools may yield inconsistent results, and some benchmarking tools may exhibit biases when developed by the message queue developers themselves. Furthermore, our benchmarking process generated comprehensive results encompassing various metrics. These metrics include throughput, measured in both the number of events per second and megabytes per second. Additionally, we evaluated the end-to-end latency across different percentiles, such as the 50th, 75th, 95th, and 99th percentile. To enhance data usability, our results were generated in JSON format, facilitating the representation of data in diverse forms for further analysis and interpretation.

The flow diagram of the methodology is shown in Figure 1.

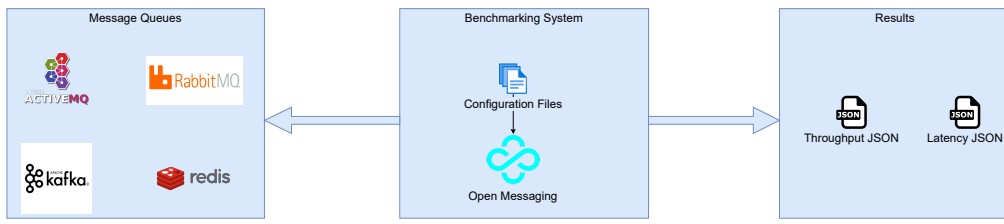

**Figure 1.** Flow diagram of the methodology.

## 5. Experiments

All of the experiments were conducted using a 2020 MacBook Air equipped with an Apple M1 chip, 16 GB of RAM, and a 512 GB SSD. In all four technologies, the experiments were carried out for a single producer and a single consumer.

In order to accurately measure the throughput of each message queue, we conducted a total of 18 experiments. For each message queue, we ran the experiment 6 times, with varying message sizes of 10 B, 100 B, 1 KB, 10 KB, 50 KB, and 100 KB. Each experiment was run for a duration of 5 min. This approach allowed us to capture a comprehensive set of data points for each message queue under different conditions, ensuring that we can evaluate their performance more accurately. To capture the throughput data, we recorded both the megabytes per second (MBps) and the number of events per second. The throughput in megabytes per second was calculated by multiplying the maximum number of events captured by the message size used. After collecting the data for each message queue, we then plotted collective graphs that show the maximum throughput in terms of megabytes per second and the maximum throughput in terms of the number of events per second.

To measure the latency, we conducted a total of 15 experiments. For each message queue, we ran the experiment 5 times, with varying throughputs of 1MBps, 3MBps, 6MBps, 10MBps, and 15MBps. In order to achieve these throughputs, we used the same data size of 32KB but varied the number of events per second. To achieve 1MBps, 3MBps, 6MBps, 10MBps, and 15MBps, we set the number of events per second to 32, 96, 192, 320, and 480, respectively. Each experiment was run for 5 min, providing ample time to collect reliable latency data. We recorded the data for the end-to-end latency for the 50th (median latency), 75th, 95th, and 99th percentile. Finally, we plotted four different latency graphs that merged the data from all four message queues, providing a comprehensive overview of latency performance across different percentiles and message queues.

## 6. Results

The results of the experiments can be divided into two subcategories: throughput and latency. The results for each subcategory are shown in their own subsection below. For ease of comparison, a graph is provided, enabling a visual assessment of the four message queues. Additionally, for precise information, the exact values are displayed in a table located below the graph, providing concrete data. The y-axis in all of the graphs has been scaled logarithmically so as to compensate for the huge difference between some of the values. For consistency, we have used the same colors to represent each of the message queues across all of the graphs: green for ActiveMQ Artemis, yellow for RabbitMQ, blue for Redis, and red for Apache Kafka.

### 6.1. Throughput

The comprehensive results of the experiments for throughput are shown in Figures 2 and 3. Figure 2 shows the maximum throughput in terms of the number of events per second whereas Figure 3 shows the maximum throughput in terms of megabytes per second.

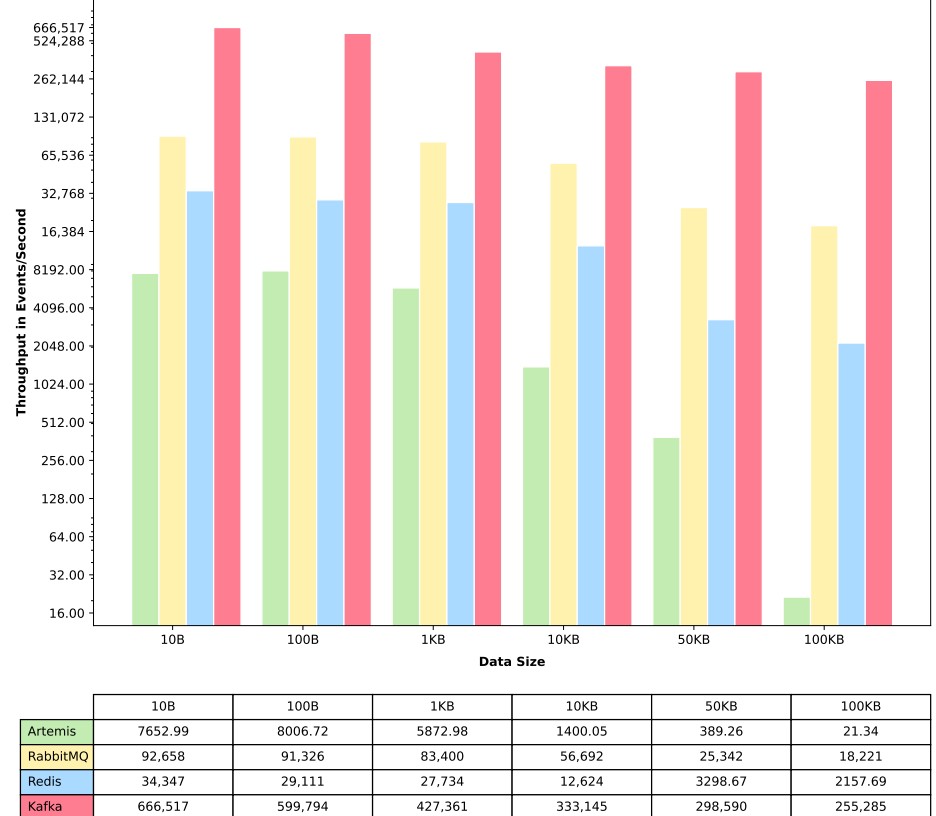

| | 10B | 100B | 1KB | 10KB | 50KB | 100KB |
|---|---|---|---|---|---|---|
| Artemis | 7652.99 | 8006.72 | 5872.98 | 1400.05 | 389.26 | 21.34 |
| RabbitMQ | 92,658 | 91,326 | 83,400 | 56,692 | 25,342 | 18,221 |
| Redis | 34,347 | 29,111 | 27,734 | 12,624 | 3298.67 | 2157.69 |
| Kafka | 666,517 | 599,794 | 427,361 | 333,145 | 298,590 | 255,285 |

**Figure 2.** Throughput in terms of events per second.

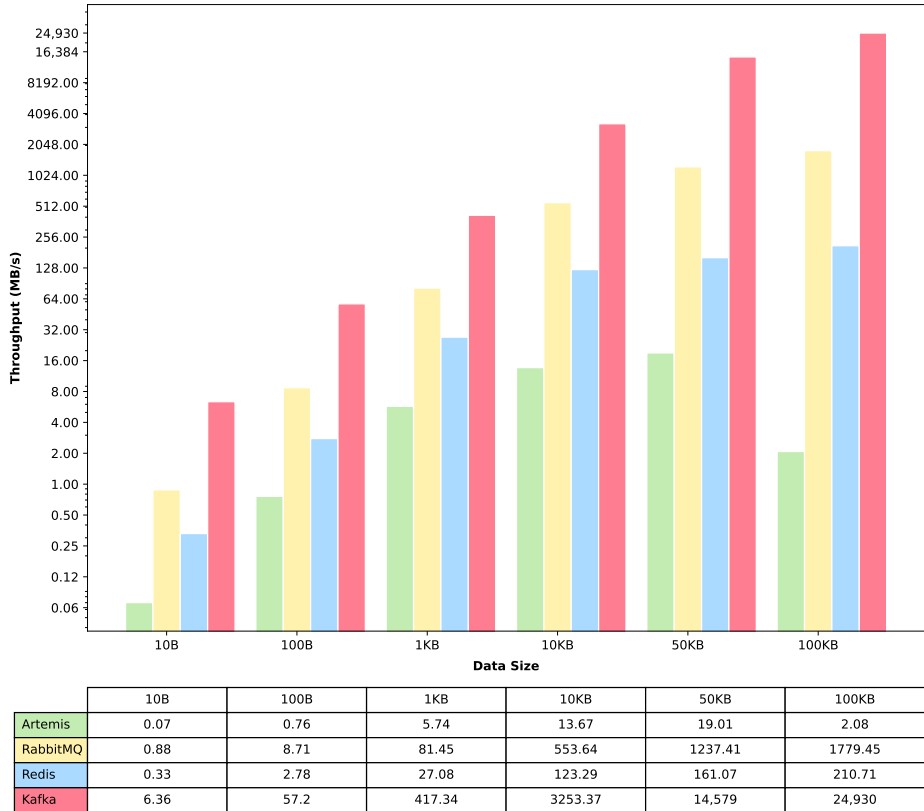

| | 10B | 100B | 1KB | 10KB | 50KB | 100KB |
|---|---|---|---|---|---|---|
| Artemis | 0.07 | 0.76 | 5.74 | 13.67 | 19.01 | 2.08 |
| RabbitMQ | 0.88 | 8.71 | 81.45 | 553.64 | 1237.41 | 1779.45 |
| Redis | 0.33 | 2.78 | 27.08 | 123.29 | 161.07 | 210.71 |
| Kafka | 6.36 | 57.2 | 417.34 | 3253.37 | 14,579 | 24,930 |

**Figure 3.** Throughput in terms of megabytes per second.

When it comes to measuring the throughput of message queues in terms of the number of events processed per second, Apache Kafka emerged as the clear winner among the four technologies. It demonstrated a remarkable performance, leaving the other three systems far behind. RabbitMQ, on the other hand, secured the second spot, but the gap between Apache Kafka and RabbitMQ was still significant. Redis secured the fourth spot. Finally, ActiveMQ Artemis ranked fourth with an even larger difference in performance when compared to the top three. A common trait noticed between all four technologies was that the number of events per second decreased when increasing the message size.

The results regarding which message queue performed the best and worst were the same for throughput in terms of megabytes per second and throughput in terms of the number of events per second. This is expected since the throughput in megabytes per second is just another method of representing throughput, as these data were calculated by multiplying the maximum number of events by the message size. However, having throughput data in these two forms could be beneficial for the reasons stated in the Methodology section. We also noticed that the throughput increases with respect to an increasing message size for Apache Kafka, RabbitMQ, and Redis. However, for ActiveMQ Artemis, when increasing the message size, the throughput increases up to a certain point and then starts to drop.

To summarize the results, Apache Kafka outperformed the other three message queues in terms of throughput by a significant margin. While RabbitMQ came in second place, the gap between RabbitMQ and Apache Kafka was still substantial. Redis lagged behind both Apache Kafka and RabbitMQ. ActiveMQ Artemis had the lowest throughput and lagged significantly behind the other message queues.

*6.2. Latency*

Figures 4–7 show the end-to-end latency for the 50th, 75th, 95th, and 99th percentiles for the throughputs of 1 MBps, 3 MBps, 6 Mbps, 10 MBps, and 15 Mbps. Based on the data presented in the four graphs, it is evident that Redis outperformed all other systems across all the tested throughputs. This is attributed to Redis storing data in memory instead of on a disk, resulting in faster data access times. Apache Kafka slightly lagged behind Redis. As we can see from the 50th percentile end-to-end latency graph, the difference between the latency of Redis and Apache Kafka is only 3 ms. On the other hand, ActiveMQ Artemis showed a better performance than RabbitMQ for lower throughputs. However, when the throughput was increased, the latency for ActiveMQ Artemis increased significantly, and was even higher than RabbitMQ. It is worth noting that Redis, Apache Kafka, and RabbitMQ demonstrated consistent latency across all the throughputs tested. In contrast, the latency for ActiveMQ Artemis seemed to steadily increase with each subsequent increase in throughput.

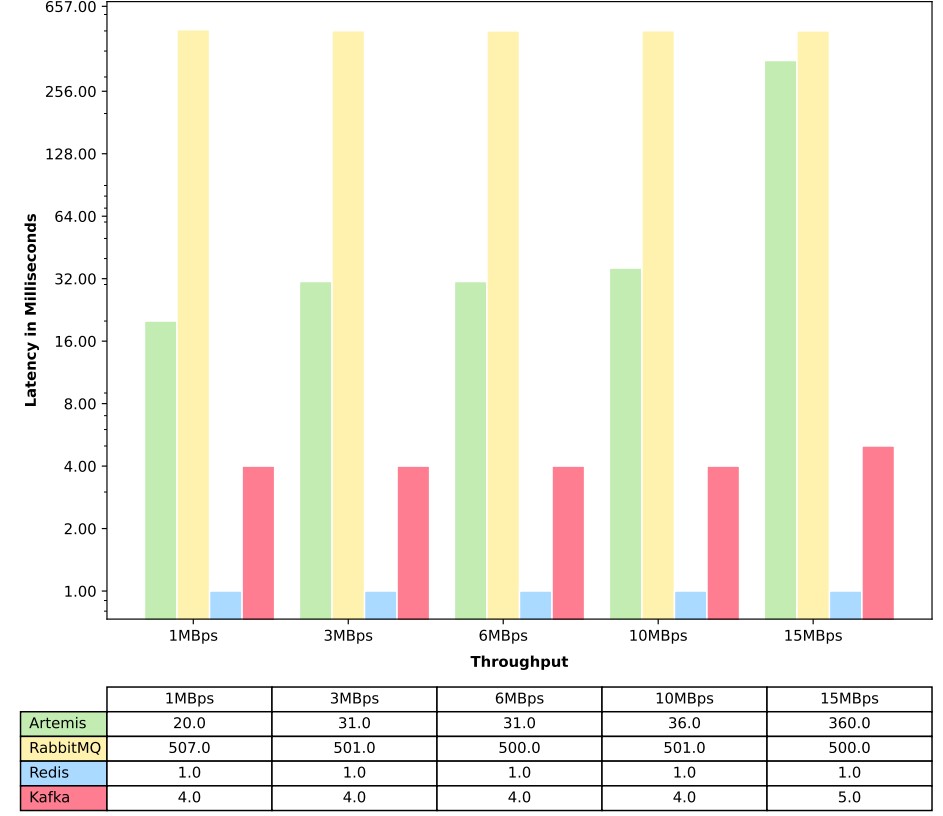

| | 1MBps | 3MBps | 6MBps | 10MBps | 15MBps |
|---|---|---|---|---|---|
| Artemis | 20.0 | 31.0 | 31.0 | 36.0 | 360.0 |
| RabbitMQ | 507.0 | 501.0 | 500.0 | 501.0 | 500.0 |
| Redis | 1.0 | 1.0 | 1.0 | 1.0 | 1.0 |
| Kafka | 4.0 | 4.0 | 4.0 | 4.0 | 5.0 |

**Figure 4.** The 50th percentile end-to-end latency.

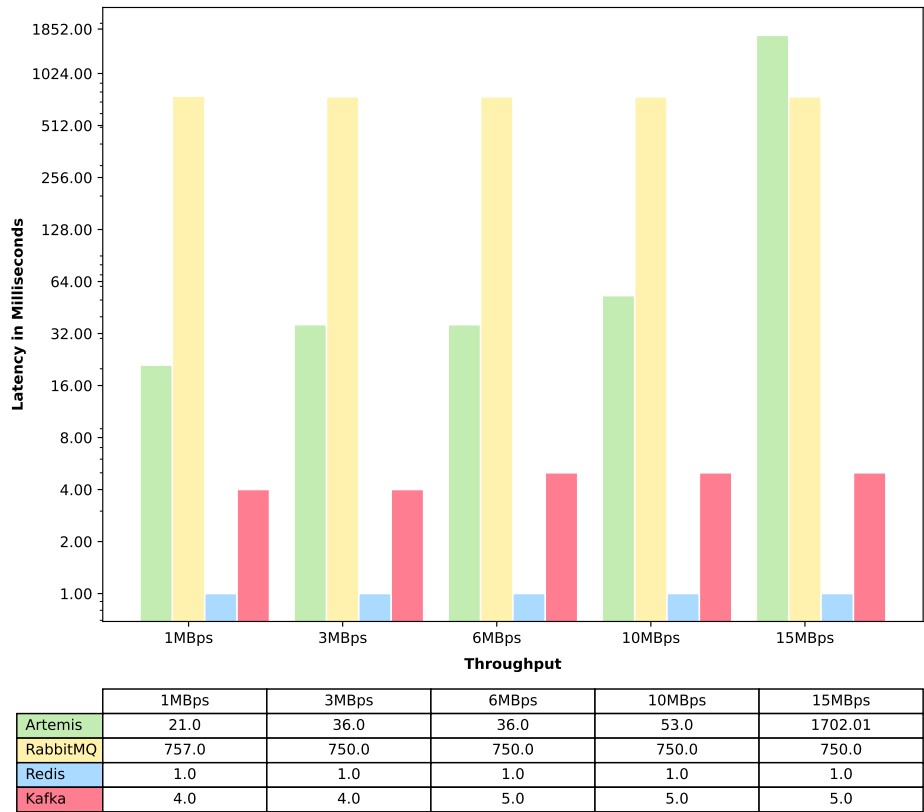

| | 1MBps | 3MBps | 6MBps | 10MBps | 15MBps |
|---|---|---|---|---|---|
| Artemis | 21.0 | 36.0 | 36.0 | 53.0 | 1702.01 |
| RabbitMQ | 757.0 | 750.0 | 750.0 | 750.0 | 750.0 |
| Redis | 1.0 | 1.0 | 1.0 | 1.0 | 1.0 |
| Kafka | 4.0 | 4.0 | 5.0 | 5.0 | 5.0 |

**Figure 5.** The 75th percentile end-to-end latency.

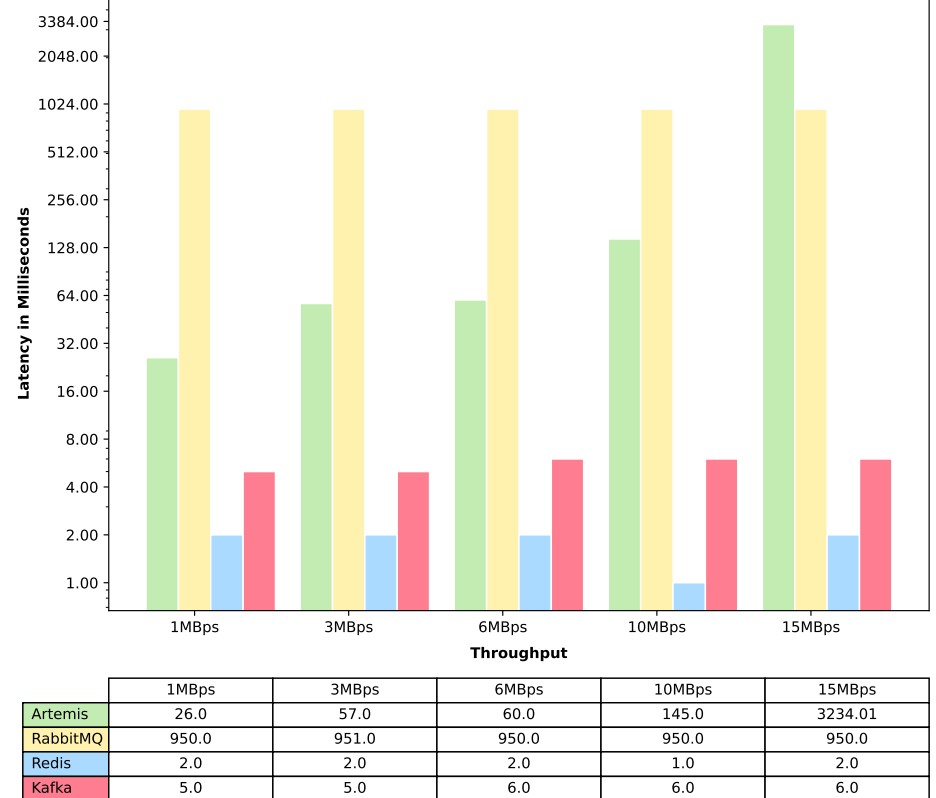

| | 1MBps | 3MBps | 6MBps | 10MBps | 15MBps |
|---|---|---|---|---|---|
| Artemis | 26.0 | 57.0 | 60.0 | 145.0 | 3234.01 |
| RabbitMQ | 950.0 | 951.0 | 950.0 | 950.0 | 950.0 |
| Redis | 2.0 | 2.0 | 2.0 | 1.0 | 2.0 |
| Kafka | 5.0 | 5.0 | 6.0 | 6.0 | 6.0 |

**Figure 6.** The 95th percentile end-to-end latency.

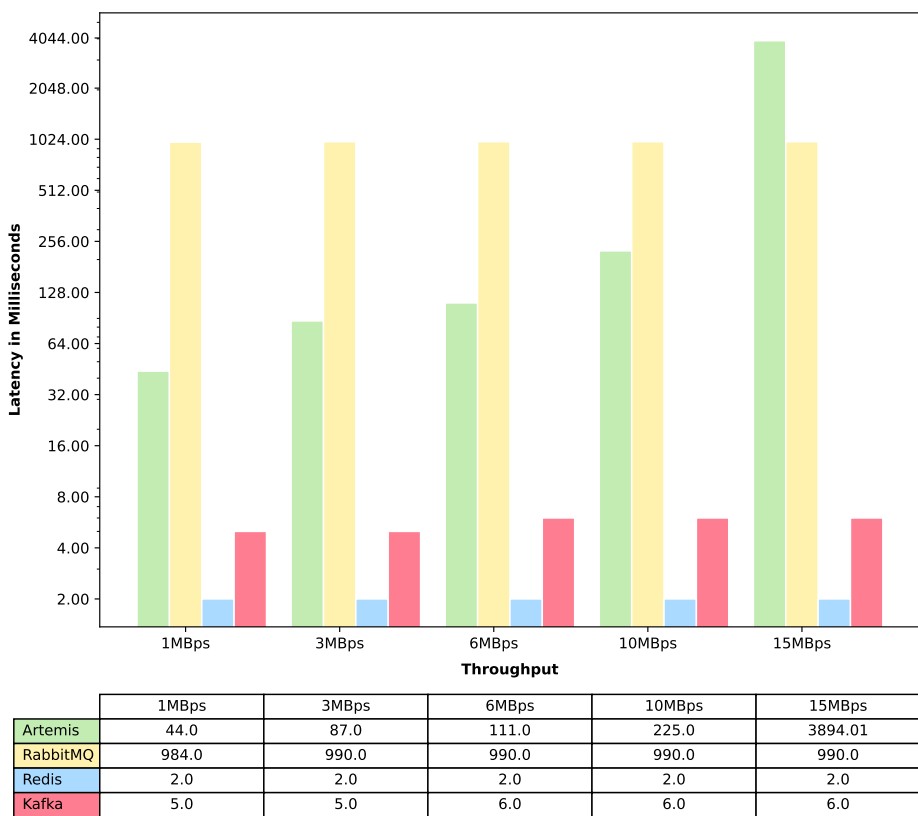

| | 1MBps | 3MBps | 6MBps | 10MBps | 15MBps |
|---|---|---|---|---|---|
| Artemis | 44.0 | 87.0 | 111.0 | 225.0 | 3894.01 |
| RabbitMQ | 984.0 | 990.0 | 990.0 | 990.0 | 990.0 |
| Redis | 2.0 | 2.0 | 2.0 | 2.0 | 2.0 |
| Kafka | 5.0 | 5.0 | 6.0 | 6.0 | 6.0 |

**Figure 7.** The 99th percentile end-to-end latency.

## 7. Discussion

Our study fairly compared the four message queues using a benchmarking tool created by a well-respected organization. We found that the different message queues have their own advantages and drawbacks. One message queue that performs well in a certain scenario might not perform well in another. To give an example of this, Redis performed the best in terms of latency. However, Apache Kafka had a significantly higher throughput than Redis. It is up to the developer or the system architect to decide which technology best suits their specific use case.

The results of this study can be used as a guideline to make an informed decision on when to use a specific technology. Below, we have outlined four different use cases and the technologies best suited for them.

### 7.1. High Throughput

If throughput is the most important to us, Apache Kafka is the technology that should be chosen. Our study showed that it significantly outperforms the other three message queues when it comes to throughput both in terms of the number of messages per second and the total size of the data. Due to its capability in achieving a high throughput, it can even be used as a data stream.

### 7.2. Latency

If our use case requires us to have a very low latency, Redis would be the obvious choice as our results show that Redis far outshines the other two technologies in terms of latency. Although we did not consider persistence in this study, it should be noted that Redis is mostly an in-memory data store with limited persistence features.

### 7.3. Low Throughput and Low Latency

In cases when storing data in memory is not an option, it is wiser to go with either ActiveMQ Artemis or RabbitMQ. Additionally, if we are certain that we will not require a high throughput, ActiveMQ Artemis would make a very good option since its latency is lower than RabbitMQ for low-throughput scenarios.

### 7.4. Low Latency and High Throughput

Redis performed the best in terms of latency. Even though Apache Kafka lagged behind Redis slightly in terms of latency, the difference was not significant. On the other hand, Apache Kafka significantly outperformed the other three message queues in terms of throughput. Hence, for this scenario, Apache Kafka would be the optimal choice.

## 8. Conclusions and Future Works

In this study, we conducted an extensive benchmarking analysis to evaluate the performance of four popular message queue systems: Redis, ActiveMQ Artemis, RabbitMQ, and Apache Kafka. Our goal was to gain deep insights into the strengths, limitations, and relative performance of each technology. To ensure consistent and unbiased evaluations, we employed the OpenMessaging Benchmark tool developed by The Linux Foundation. We focused on two key metrics: latency and throughput, to compare the message queue systems. The results of our study provide valuable insights for practitioners working with message queues.

In terms of latency, Redis emerged as the top performer, exhibiting an exceptional performance with consistently low latency. Apache Kafka closely followed Redis, with a negligible difference of approximately 4 milliseconds. ActiveMQ Artemis outperformed RabbitMQ for lower-throughput scenarios, showcasing its strengths in specific use cases. However, as the throughput increased, the latency of ActiveMQ Artemis escalated significantly, surpassing that of RabbitMQ.

Regarding throughput, Apache Kafka demonstrated remarkable capabilities, securing the first position by a considerable margin. RabbitMQ secured the second spot, displaying

a commendable throughput performance. Redis followed closely behind, although its throughput capabilities were comparatively lower than RabbitMQ and Apache Kafka. ActiveMQ Artemis performed the least efficiently in terms of throughput, highlighting its limitations in handling high-volume message traffic.

These findings provide practitioners with valuable guidance in selecting the most suitable message queue system based on their specific requirements. It is important to consider not only the raw performance metrics but also the trade-offs and considerations associated with each technology. Additional factors such as ease of use, community support, and integration capabilities should also be taken into account to make informed decisions.

In conclusion, this benchmarking study contributes to the body of knowledge in the field of message queue systems by providing a comprehensive evaluation of Redis, ActiveMQ Artemis, RabbitMQ, and Apache Kafka. The insights gained from this research can guide practitioners in selecting the most appropriate message queue technology for their use cases, considering factors such as latency, throughput, and other relevant considerations.

In future research, it would be valuable to expand upon the findings presented in this paper. Firstly, exploring additional performance metrics beyond latency and throughput would provide a more comprehensive understanding of the strengths and weaknesses of Redis, ActiveMQ Artemis, RabbitMQ, and Apache Kafka. This could include factors such as persistence, scalability, fault tolerance, and resource utilization. Additionally, conducting experiments with larger and more diverse workloads, spanning various use case scenarios, would further validate and refine the performance comparisons. Moreover, investigating the impact of different deployment architectures, such as cloud-based or containerized environments, on the message queues' performance would provide insights into their adaptability and suitability in modern computing infrastructures. Lastly, considering the integration and interoperability aspects of these message queues with other technologies and frameworks commonly used in distributed systems would be beneficial for practitioners seeking to leverage them in real-world applications.

**Author Contributions:** Conceptualization, R.M. and T.C.; methodology, R.M.; software, R.M. and M.S.H.C.; validation, R.M.; formal analysis, R.M. and M.S.H.C.; investigation, R.M.; resources, R.M.; data curation, R.M. and M.S.H.C.; writing—original draft preparation, R.M. and M.A.A.; writing—review and editing, R.M., M.A.A. and T.C.; visualization, R.M. and M.S.H.C.; supervision, R.M. and T.C.; project administration, R.M. and T.C.; funding acquisition, T.C. All authors have read and agreed to the published version of the manuscript.

**Funding:** This research was funded by the National Science Foundation under grant no. 1854049 and a grant from Red Hat Research https://research.redhat.com (accessed on 7 June 2023).

**Data Availability Statement:** The results in JSON format can be found under results folder of the GitHub repository https://github.com/rokinmaharjan/openmessaging/tree/master/results (accessed on 7 June 2023).

**Conflicts of Interest:** The authors declare no conflict of interest.

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
