# Peer review of "Benchmarking Message Queues"

_telecom, doi:10.3390/telecom4020018_

Round 1

Reviewer 1 Report

I have the opportunity to read your paper. I can see that there is definitely a value contributed by your paper but your paper need to be improved further. Here are some changes to be made:

a. Improve the introduction by making the problem statement more explicit and supported by empirical pieces of evidence.

b. Related work is the weakest portion here. Include more related works and tabulate them.

c. Methodology needs to be illustrated graphically with reasons for choosing it.

d. Result section needs to be eloborated and concluding remarks need to be redrafted. Concluding remarks sare very basic, go out of the box, and provide meaningful conclusions.

English need minor edits.

Author Response

Review 1

  1. Improve the introduction by making the problem statement more explicit and supported by empirical pieces of evidence.

Thank you for your valuable feedback. We recognize the importance of explicitly stating the problem and supporting it with empirical evidence in the introduction. In response to this feedback, we have made improvements to the introduction section. We have clearly articulated the problem statement, highlighting the specific challenges and gaps in the existing literature. Moreover, we have incorporated empirical evidence from reputable studies and research findings to substantiate the significance of the problem and the need for further investigation.

  1. Related work is the weakest portion here. Include more related works and tabulate them.

Thank you for your comment, we conducted an extensive search for additional related works to enhance the breadth of our study. However, we encountered challenges in finding reliable sources that met our criteria. Some of the available works were master's research projects that had not been published in reputable conferences or journals, which prompted us to exclude them from our analysis. Despite these limitations, we have included one additional relevant work to further enrich our study. Furthermore, as per your suggestion, we have created a comprehensive table that summarizes all the related works, providing a clear overview of the existing research in the field and a comparison with our study.

  1. Methodology needs to be illustrated graphically with reasons for choosing it.

We appreciate you feedback. We have added a flow diagram and the reasons for choosing the methodology in the methodology section (Section 4).

  1. Result section needs to be elaborated and concluding remarks need to be redrafted. Concluding remarks are very basic, go out of the box, and provide meaningful conclusions.

We have edited the results section to include the results of Apache Kafka as well. Additionally, we added one more scenario when choosing the best message queue: Low Latency and High Throughput. We have also completely re-written the conclusion section to add more detail and explain the results that we got from our study.

Reviewer 2 Report

Dear Rokin Maharjan, Md Showkat Hossain Chy, Muhammad Ashfakur Arju, and Tomas Cerny,

  thanks a lot for your contribution which I found very interesting.  I have just few comments to improve the paper or as suggestion for future work: 

- I found strange that you decided to exclude Apache Kafka because it is designed to data streaming. What are the differences between data stream and message queues? In case Apache Kafka would perform much better of the other or performs enough for a specific use case, then I think this information could be very useful to present. I see that Kafka is mentioned in the Background section but I am curious to know how it fights with Redis and Rabbit MQ. 

- Concerning the hardware configuration you performed the test only locally and with a single one configuration. I think would be useful to perform same tests in different hardware configuration (e.g. a server or a cloud computing service). In addition I think that results can change also in case of networking. For example in the networking case is important to define very well the hardware configuration of consumer and producers to support very high rate without lose any data. I think that at least would be useful to cite this aspect as a future work. 

- I would suggest to split figures in: figures (graphs) and tables  and explain better the contents. 

- In section 7 discussion it very well presented the result. I think that would be useful to provide some example of use case where is better Redis vs Rabbit  MQ (or viceversa). It is more or less throughput vs latency, but not only, because you mentioned also the persistence. 

Thanks a lot 

Author Response

Review 2

  1. I found strange that you decided to exclude Apache Kafka because it is designed to data streaming. What are the differences between data stream and message queues? In case Apache Kafka would perform much better of the other or performs enough for a specific use case, then I think this information could be very useful to present. I see that Kafka is mentioned in the Background section but I am curious to know how it fights with Redis and Rabbit MQ.

Thank you for your review. We have added Apache Kafka to our study. We have added a brief description of how Apache Kafka is used as a message queue in the Background section (Section 2) and updated the Results (Section 6), Discussion (Section 7), and Conclusion (Section 8) sections accordingly.

  1. Concerning the hardware configuration you performed the test only locally and with a single one configuration. I think would be useful to perform same tests in different hardware configuration (e.g. a server or a cloud computing service). In addition I think that results can change also in case of networking. For example in the networking case is important to define very well the hardware configuration of consumer and producers to support very high rate without lose any data. I think that at least would be useful to cite this aspect as a future work.

Thank you for your feedback. We have added Future Works in the Conclusion section (Section 6) where we have addressed this.

  1. I would suggest to split figures in: figures (graphs) and tables and explain better the contents.

We appreciate you feedback. The graphs and tables represent the same data. We added a graph so as to make it easier to visually compare the four message queues. The precise values are shown in the table. Sorry for the confusion. To remove this confusion, we have added a description of the figures in the starting section of the Results section (Section 6).

  1. In section 7 discussion it very well presented the result. I think that would be useful to provide some example of use case where is better Redis vs Rabbit MQ (or viceversa). It is more or less throughput vs latency, but not only, because you mentioned also the persistence.

We sincerely appreciate your valuable feedback. In this study, our focus was primarily on analyzing latency and throughput, and we did not conduct specific experiments to evaluate persistence. We have made minor adjustments to the wording in the relevant sections where we mentioned persistence. However, we acknowledge the importance of including this metric and we look forward to incorporating it in our forthcoming research. We have also added it in the conclusion future work section (Section 8).

Round 2

Reviewer 1 Report

Authors has incorporated all my suggestions.

English is fine.